# Polychrony as Chinampas

Eric Dolores-Cuenca [1], José Antonio Arciniega-Nevárez [2,*], Anh Nguyen [3], Amanda Yitong Zou [4], Luke Van Popering [5], Nathan Crock [5,6], Gordon Erlebacher [6] and Jose L. Mendoza-Cortes [7,*]

1. Department of Mathematics, Yonsei University, Seoul 03722, Republic of Korea
2. División de Ingenierías, Campus Guanajuato, Universidad de Guanajuato, Guanajuato 36000, Mexico
3. College of Arts and Sciences, Drexel University, Philadelphia, PA 19104, USA
4. Department of Mathematics, University of Michigan, Ann Arbor, MI 48104, USA
5. Emelex, Tallahassee, FL 32308, USA
6. Department of Scientific Computing, Florida State University, Tallahassee, FL 32306, USA
7. Department of Chemical Engineering & Materials Science, Michigan State University, East Lansing, MI 48824, USA
* Correspondence: ja.arciniega@ugto.mx (J.A.A.-N.); jmendoza@msu.edu (J.L.M.-C.)

**Abstract:** In this paper, we study the flow of signals through linear paths with the nonlinear condition that a node emits a signal when it receives external stimuli or when two incoming signals from other nodes arrive coincidentally with a combined amplitude above a fixed threshold. Sets of such nodes form a polychrony group and can sometimes lead to cascades. In the context of this work, cascades are polychrony groups in which the number of nodes activated as a consequence of other nodes is greater than the number of externally activated nodes. The difference between these two numbers is the so-called profit. Given the initial conditions, we predict the conditions for a vertex to activate at a prescribed time and provide an algorithm to efficiently reconstruct a cascade. We develop a dictionary between polychrony groups and graph theory. We call the graph corresponding to a cascade a chinampa. This link leads to a topological classification of chinampas. We enumerate the chinampas of profits zero and one and the description of a family of chinampas isomorphic to a family of partially ordered sets, which implies that the enumeration problem of this family is equivalent to computing the Stanley-order polynomials of those partially ordered sets.

**Keywords:** polychrony; nonlinear signal flow graph; cellular automata; rule 192; order polynomials; Ehrhart series

## 1. Introduction

Networks directly or indirectly impact many aspects of our lives via numerous modalities, including the internet, telecommunications, social media, the brain, and our bodies. These networks can be modeled through signal flow graphs, or directed graphs in which nodes represent system variables and the edges represent functional connections between pairs of nodes. In this paper, we investigate particular examples of nonlinear signal flow graphs, see Section 2 in which some external stimuli are applied to the vertices of the graph, triggering a chain reaction on these and other vertices.

A cascade refers to those chain reactions in which the number of external stimuli is smaller than the number of reactions generated within the chain. The time duration of these stimuli plays an important role, rendering the study of cascades nonlinear.

A vertex subjected to an external stimulus or triggered indirectly as a consequence of reactions to the stimuli of others is considered activated. In this context, we seek answers to the following questions:

- Is a given vertex activated at a particular time?
- Can we reconstruct all the vertices that will change their states to activated?



Neuronal networks in which the flow of signals form time-locked patterns define the polychrony groups [1]. The word polychrony is derived from the Greek words "poly" (i.e., many) and "chronous" (i.e., time).

Most of the literature focuses on linear signal flow graphs, where the edges represent matrix operations. However, studying polychrony groups requires a language to treat the nonlinear case. As part of our methodology, we adopt the language of graph theory to model signal flow networks. A polychronous group without redundant information is encoded in a graph called a chinampa. The study of chinampas is described in Sections 3–5. Section 3 describes the signal flow networks from the point of view of graph theory. In Section 4, we provide a topological characterization of chinampas (see Theorem 1). Section 5 explains the relationship of our work with cellular automata.

We introduce the concept of profit, a measure of how many vertices activate in response to external stimuli. In Section 6, we give formulae for the number of pyramids in a chinampa of profits zero and one. In Section 6.3, we provide the code to answer both of our target questions. The algorithm we implemented is available at https://github.com/mendozacortesgroup/chinampas/ (accessed on 3 March 2023). Note that the algorithm works with an input network of the form $1 \to 2, \cdots, \to n$ and when the input network is a tree. Our code is optimal compared with the state of the art, as explained in the conclusions.

The study of chinampas of a profit greater than one is more difficult. In Section 7, we describe a family of chinampas whose enumeration problem is equivalent to computing the order polynomial of some posets. The order polynomial counts the labeling maps from a poset to chains $1 < 2 < \cdots < n$, and the study of order polynomials is an active area of research in enumerative combinatorics. Our work sets the basis for the study of polychrony groups in combinatorics.

## 2. Nonlinear Signal Flow Graphs

In this article, we shall study a kind of *nonlinear signal flow graph* called a directed graph. We think that a signal propagates throughout an edge by following its orientation. When several signals reach a vertex simultaneously, a built-in condition called the *threshold* determines whether it will react by firing signals. The concept of the signal flow graphs was developed by Samuel Mason and Claude Shannon [2,3]. If the condition is linear in the intensity of the input signals, then the graphs are called *linear signal flow graphs*.

We study the following *nonlinear* condition of a signal flow graph:

- Every signal has an intensity of one.
- Every vertex has a threshold intensity of two.
- If a vertex coincidentally receives signals of an intensity higher or equal to the threshold, then the vertex fires a signal through each of its outgoing edges.

Nonlinear signal flow graphs are used to study circulatory regulation [4], to design automatization of nonlinear data converters [5], to compare system-level and spice-level static nonlinear circuits [6], to build models for DC-DC buck-boost converters [7], and to analyze the problem of inverting a system consisting of nonlinear and time-varying components [8]. The nonlinear condition of our signal flow graphs is an abstraction of neural spikes. Neural spikes are used in cognitive computing to develop hardware that emulates the human nervous system [9–13], to implement robust chaotic communication [14], and to power efficient channel coding [15]. The link between neural spikes and pulse position modulation is explained in [16].

In particular, we are interested in polychrony groups defined as a group of *primary neurons* (vertices) that fire at specific times, leading to *secondary neurons* firing. As a result, a *cascade* occurs when the number of primary neurons is below the number of secondary neurons (see [1] for more details).

We study the general phenomena of polychronization of nonlinear signal flow graphs of the form

$$1 \to 2 \to 3 \to \cdots \to n.$$

Our initial objective is to characterize the polychrony groups that lead to cascades. A second goal is to count the families of the cascades. We also explain the algebraic structure of cascades using cellular automata theory. Our final goal is to develop an algorithm that can answer certain types of queries without the need to compute all possible interactions. An example of such a query might be establishing which neuron (vertex) will be activated at some future time.

### 3. Base and Activation Diagrams

In this section, we introduce the notion of a covering graph to represent the flow of signals in a network. As a part of our methodology, we translate the problem of characterizing our nonlinear signal flow graphs into the problem of color-covering graphs.

#### 3.1. Base Diagram

Consider the set of nonnegative integers $\mathbb{N}$. For a directed graph $A$, we denote the set of vertices and edges of $A$ as $V(A)$ and $E(A)$, respectively. We define a *base diagram* of $A$ as the pair $(B, p)$ where $B$ is a directed graph with vertices $V(B) = V \times \mathbb{N}$ and, given any edge $m \to n$ of $E(A)$ labeled by $t \in \mathbb{N}$, we define an edge in $E(B)$ as $(m, i) \to (n, i + t)$ between the vertices $(m, i)$ and $(n, i + t)$ of $V(B)$. The coordinate $i$ in $(m, i)$ indicates the row position of the vertex (see Figure 1).

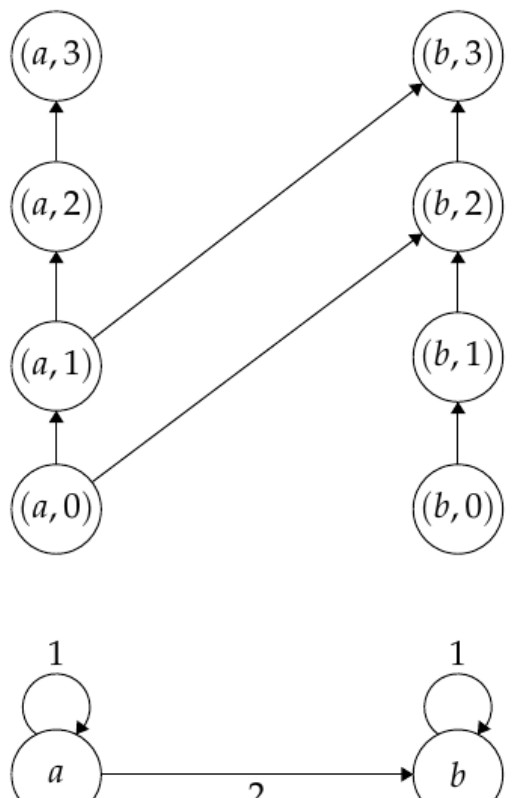

**Figure 1.** In the bottom part of the figure, a simple graph $A$ is shown. In the top part, the corresponding base diagram $B$ is shown.

The function $p : B \to A$ is the projection defined by $(m, i) \mapsto m$, and the image of every edge $(m, i) \to (n, i + t)$ under $p$ defines the edge $m \to n$ of $E(A)$ with a label $t$.

We are interested in particular types of directed graphs. A *singleton u* is a graph with one vertex and one self-edge with a label 1. A *path of length l*, denoted by $path(l)$, is a directed graph with vertices $\{1, 2, \cdots, l\}$, where each vertex has a self-edge, while each vertex $i < l$ has one outgoing edge to the vertex $i + 1$. Any edge has a label 1. A *cycle of length l*, denoted by $cyc(l)$, is a path with vertices in the set $\{1, 2, \cdots, l\}$, but the vertex $l$ has

one outgoing edge to vertex 1 with a label 1. In other words, a cycle is a closed path. The graphs *u*, *path(l)*, and *cyc(l)* have their own base diagrams, which we call *ũ*, *tellis(l)*, and *cylinder(l)*, respectively. Figure 2 shows examples of a path, a cycle, and their respective base diagrams.

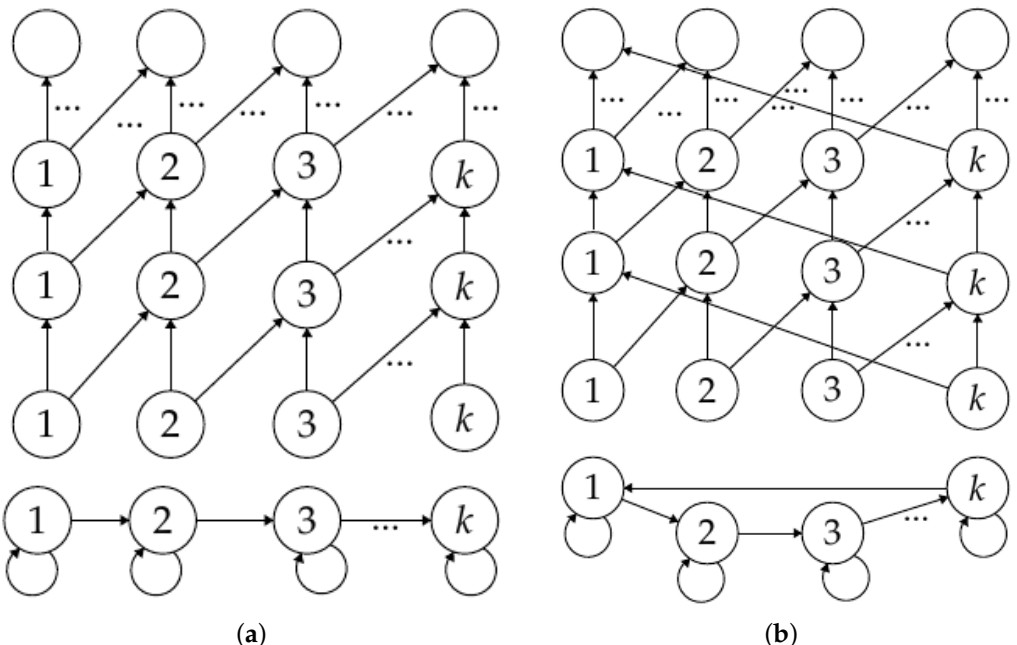

**Figure 2.** Directed labeled graphs and their correponding base diagrams. (**a**) A path(*l*) and its corresponding base diagram *tellis(l)*. (**b**) A *cyc(l)* and its corresponding base diagram cylinder(*l*).

*3.2. Activation Diagram*

Given a base diagram, we selected a subset of vertices and called them the *primary vertices*. The *activation diagram* $(B, S)$ is a base diagram $B$ and a subset $S$ of the primary vertices of $V(B)$. A *secondary vertex* with the coordinates $(r, t)$ is a vertex in the base diagram in which each one of $(r, t - 1)$ and $(r - 1, t - 1)$ is either a primary vertex or a secondary vertex. We associated the *activation graph* (i.e., the underlying colored graph in which the primary and secondary vertices are black and the remaining vertices are white) to each activation diagram. To simplify the interpretation of theactivation diagrams, we only drew vertices which were either primary or secondary and avoided the others in the base diagram (see Figure 3).

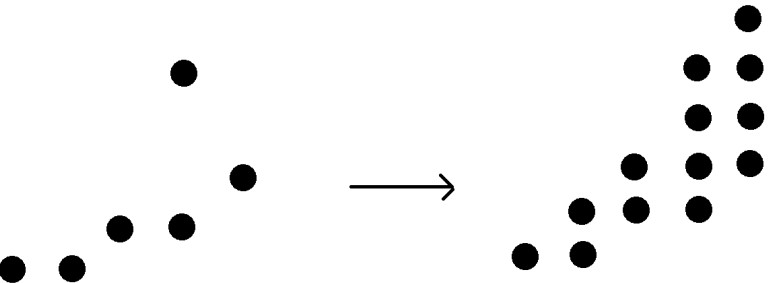

**Figure 3.** (**left**) An example of an activation diagram. The activation diagram turns into an activation graph. (**right**) A new black vertex appears in a row of the activation graph due to the two corresponding black vertices in the previous row in the activation diagram.

**Example 1.** *Consider $P = path(l)$ and its corresponding base diagram $B = tellis(l)$, as described before. Fix a vertex $v_i \in P$, and consider $S = \{(v_{i+k}, t)\} \subset B$ with $k = 0, 1, \cdots, l - 1$ consecutive vertices with the same time. The set $S$ is determined by the initial vertex $v_i$ of the path $P$, its length*

*l, and a fixed time $t \in \mathbb{N}$. We employ a* pyramid *in the activation diagram $(B, S)$ and denote it as* $pyramid(l, i, t)$. *In Figure 4, the activation diagram $pyramid(3, i, t)$ of a path with a length of three is illustrated.*

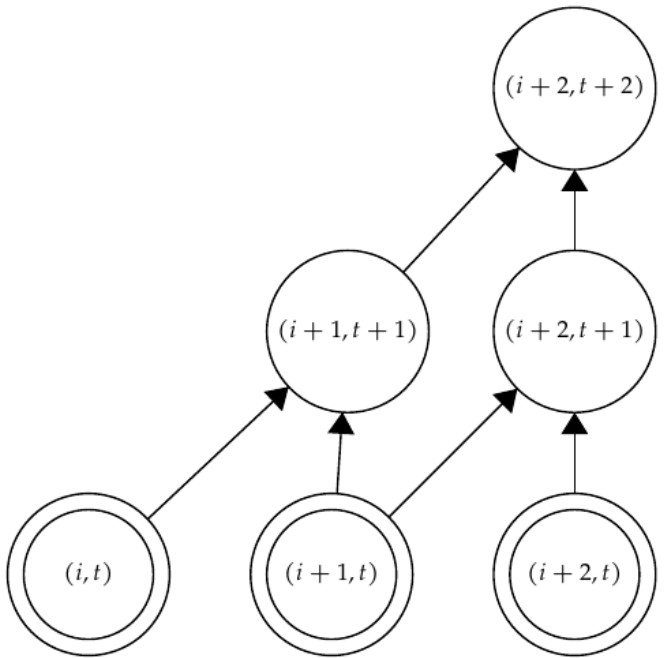

**Figure 4.** A ring (double circle) indicates the primary vertices. There are three secondary vertices, forming a pyramid.

**Remark 1.** *Suppose we have an activation diagram in which there is $pyramid(l, i, t)$ with an extra primary vertex $(i - 1 + s, t + s)$ where $0 \leqslant s \leqslant l - 1$. The vertex affects $pyramid(l, i, t)$ by activating the vertices in the diagonal $\{(i - 1 + k, t + k) \mid k = s + 1, \cdots, l\}$. On the other hand, if the extra primary vertex is localized in $(i + l + 1, t + s)$, where $0 \leqslant s \leqslant l - 1$, then the vertical line $\{(i + l + 1, t + k) \mid k = s + 1, \cdots, l\}$ is activated (see Figure 5).*

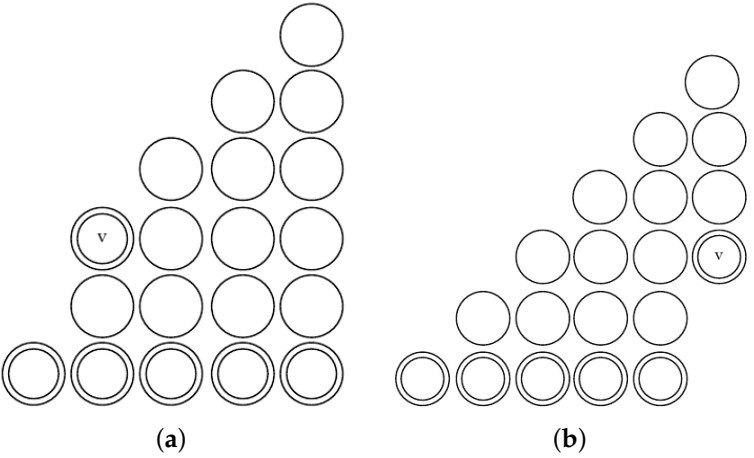

(**a**)          (**b**)

**Figure 5.** Effect of adding a primary vertex to a pyramid. (**a**) If we add an activated vertex $v$ to the left side of a pyramid, then the vertices in the same diagonal with times greater than the time of $v$ are activated until reaching the top of the pyramid. (**b**) If we add an activated vertex $v$ to the right side of the pyramid, then the vertices in the same column with a time greater than the time of $v$ are activated until reaching the top of the pyramid.

With a path of secondary vertices from $(i, t)$ to $(j, s)$, we define a sequence

$$(i_0, t_0), (i_1, t_1), \cdots, (i_k, t_k)$$

with the first point $(i_0, t_0) = (i, t)$ and final point $(i_k, t_k) = (j, s)$ such that there is an edge between two consecutive vertices. Here, the edges are considered while ignoring the direction. The edges go from $(i, t)$ to $(i, t + 1)$ and from $(i, t)$ to $(i - 1, t - 1)$.

We restricted our study to activation diagrams where every primary vertex contributed to at least one secondary vertex. A *redundant activation diagram* occurs when a primary vertex is also a secondary vertex (see Figure 6).

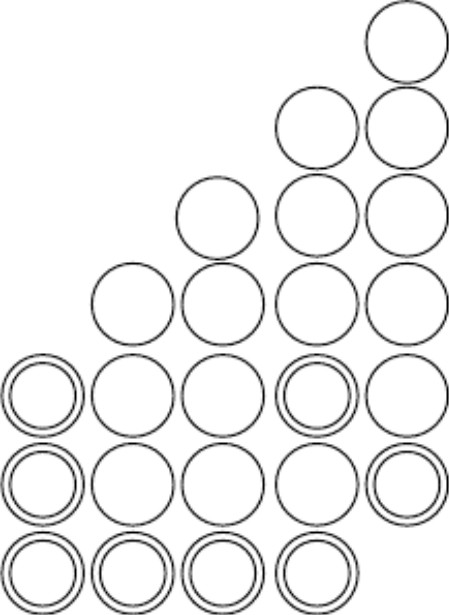

**Figure 6.** In this activation graph, there is a primary vertex which is also a secondary vertex.

An activation diagram $C$ is *connected* if the secondary vertices of the corresponding activation graph form a connected, undirected graph. As an example, Figures 3 and 4 are connected activation diagrams. Figure 7 is not a connected activation diagram.

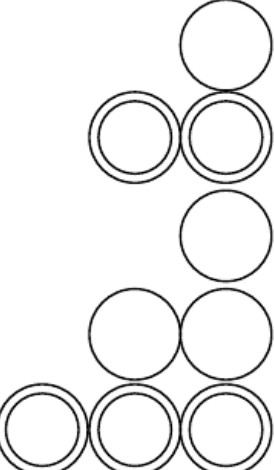

**Figure 7.** An example of a non-connected activation diagram. There is no edge between the pyramid below and the pyramid above.

## 4. Chinampas

Given an activation diagram, the *profit* is equal to the number of secondary vertices minus the number of primary vertices. The profit measures the maximum number of extra vertices which are activated as a consequence of the topology of the graph. A connected, non-redundant activation diagram is a *chinampa* if its profit is greater than or equal to zero. An example of a chinampa is illustrated in Figure 8. The simplest chinampa, in the sense that it involves the least number of vertices, is $pyramid(3, i, t)$, as shown in Figure 4. The name is due to the similarity of the figures with an ancestral Mexican agricultural technique that uses soil to grow crops on a lake. We imagine that chinampas have crops above the soil, and underneath, there are roots.

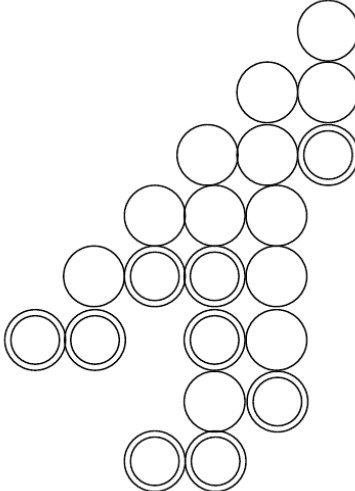

**Figure 8.** Example of a chinampa.

The profit defines a function from the set of chinampas to nonnegative integers. We denote with $profit(C)$ the profit of a chinampa $C$.

*The Topological Description of a Chinampa*

We describe chinampas over a $path(l)$. For this goal, we will give the decomposition of a chinampa into pyramids. Remember that we defined $pyramid(l, i, t)$ as an activation diagram $(B, S)$ where the primary vertices $S$ are consecutive and have the same time. We extend the definition of a pyramid to the activation diagram $(B, S)$, in which $S$ contains the secondary vertices. Thus, in chinampa $C$ of Figure 9, we have two pyramids $pyramid(3, i, t)$ and $pyramid(3, i + 2, t + 2)$, where the vertex $(i + 2, t + 2)$ is a secondary vertex.

Note that a pyramid has only one secondary vertex at the top, which we call the *pyramidion*. If $P_1 = (B, S_1)$ and $P_2 = (B, S_2)$ are pyramids in a chinampa, then we say that pyramid $P_2$ is *stacking into* pyramid $P_1$ if the pyramidion of $P_2$ is a secondary vertex of $P_1$ other than the pyramidion of $P_1$, and $P_2$ is not contained in $P_1$.

Remark 1 shows that the activation diagram of a $path(l)$ is of the form $pyramid(k, i, t)$, together with the primary vertices at the right or left of $pyramid(k, i, t)$. We will see that an activation diagram is a sequence of stacked pyramids:

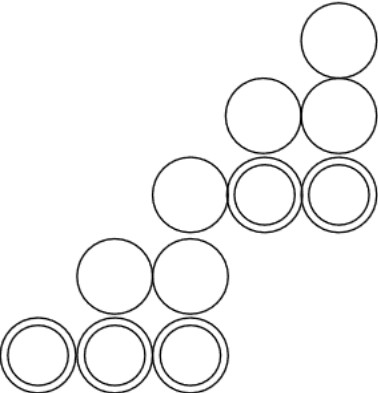

**Figure 9.** *pyramid*$(3, i, t)$ and *pyramid*$(3, i + 2, t + 2)$ with $(i + 2, t + 2)$ as a secondary vertex, forming a chinampa.

**Remark 2.** *Consider two pyramids pyramid*$(l_1, i, t)$ *and pyramid*$(l_2, i + l_1, t)$. *The pair of primary vertices* $(i + l_1, t)$ *and* $(i + l_1 + 1, t)$ *activates* $(i + l_1 + 1, t + 1)$*, which is not part of any of the two pyramids. Based on Remark 1, the entire diagonal to which the vertex belongs is activated, as well as the vertical line* $\{(i + l_1 + 1, t + k)\}$ *with* $k \in \{1, \cdots, l_1 + 1\}$*, and so on. Therefore, we end up with pyramid*$(l_1 + l_2, i, t)$*. This argument works for two pyramids: pyramid*$(l_1, i, t)$ *and pyramid*$(i + l_1 + k, i, t)$*, separated by activated vertices* $\{(i + l_1 + j, t) \mid 1 \leqslant j \leqslant k\}$*. Thus, in such cases, instead of considering several small adjacent pyramids, we always consider only the biggest pyramid that includes the small adjacent pyramids.*

**Proposition 1.** *There exists only one pyramidion with the maximum time in a chinampa. We call this a* spike*.*

**Proof.** Suppose $(i, t)$ and $(j, t)$ with $i < j$ are pyramidia with the maximum time. Through connectedness, there is a path from $(i, t)$ to $(j, t)$ with only secondary vertices. Each vertex on the path has a time lower than or equal to $t$. Starting from the vertices with lower times, we use Remark 2 until we reach those vertices with a time $t$. At each step, we conclude that all vertices above and between those in the path are secondary vertices. Then, those vertices between $(i, t)$ and $(j, t)$ are activated, meaning that $(i, t)$ and $(j, t)$ are not pyramidia, which is a contradiction. $\square$

**Remark 3.** *If* $(i, t)$ *and* $(j, t)$ *are activated vertices connected by a path of activated vertices with a time lower than or equal to $t$, then the argument in the proof of Proposition 1 shows that both of them are in a pyramid whose pyramidion has a time greater than $t$. As a consequence, if two pyramids $P_1$ and $P_2$ can be stacked under a given $P$, then pyramid $P_1$ cannot be adjacent to $P_2$, since under Remark 2, we would instead stack the biggest pyramid, which includes both $P_1$ and $P_2$. The activation diagram of Figure 10 is not a chinampa. Assume $t$ is the minimum time of the primary vertices. Then, pyramid*$(2, i, t)$ *and pyramid*$(2, i + 2, t)$ *are adjacent, and thus they are in pyramid*$(4, i, t)$*, which implies that the middle vertex becomes a secondary vertex, which is contrary to the non-redundancy requirement.*

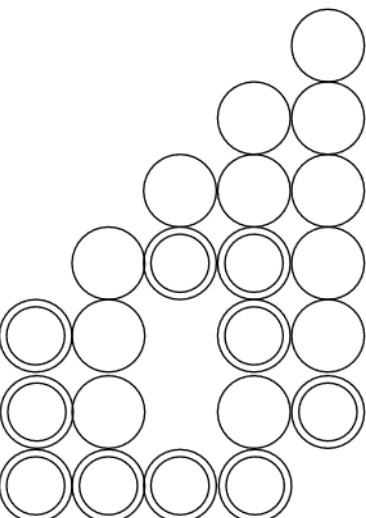

**Figure 10.** Here is an example of an activation diagram which is not a chinampa. The copies of $pyramid(2, i, t)$ at the bottom are next to each other, and as a consequence, all vertices in the hole are internally activated.

**Remark 4.** *Another consequence of Remark 1 is that a chinampa has no activated vertex located to the right of the spike. If the spike is $(i, t)$, then all activated vertices are of the form $(j, t')$ with $j \leqslant i$ and $t' < t$.*

Now, we shall give an order to the pyramids in a chinampa:

**Proposition 2.** *In a chinampa, there exists a unique $pyramid(l, i, t)$ with $l \geqslant 3$ of the maximum time. We call it the* top pyramid.

**Proof.** Start with the spike $(i, t)$. If this is the pyramidion of $pyramid(l, i, t)$ with $l \geqslant 3$, then we are done. If not, then we have a sequence of $pyramid(2, i, t)$ stacked onto each other. However, the chinampa has a profit greater than or equal to zero, so eventually, we will come across $pyramid(l, i, t)$ with $l \geqslant 3$. Let the top pyramid be the first instance found with this strategy. There is no other pyramid $pyramid(l', i', t')$ with $l' > 2, t' \geqslant t$, since through connectedness and Remark 2, we would conclude that there is a bigger pyramid containing both the top pyramid and $pyramid(l', i', t')$, which contradicts the assumption that the top pyramid is the first instance found. $\square$

**Theorem 1** (Topological classification of chinampas)**.** *Any chinampa can be described as a sequence of pyramids stacked onto each other.*

**Proof.** We showed that in a chinampa, there is a unique top pyramid (Proposition 2). This top pyramid is stacked onto a sequence of $pyramid(2, i, t)$ unless the spike belongs to the top pyramid (Proposition 1). Any other $pyramid(n, i', t')$ ($n \geqslant 3$) must have $t' < t$ (Remark 4). Pyramids are connected, which is only possible if they are stacked onto each other. $\square$

The average chinampa is described in Figure 11.

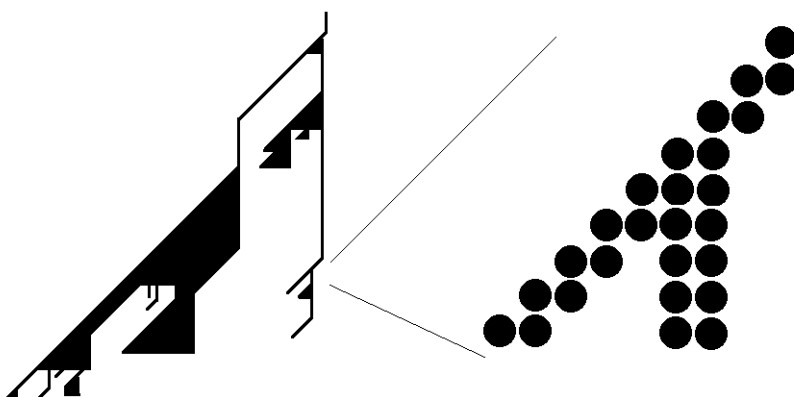

**Figure 11.** Standard example of a chinampa. (**left**) Zoomed-out image of a chinampa, where the chinampa can be represented by stacking pyramids. (**right**) Zoomed-in image of a chinampa, where we see the details of the activation diagram. The lines in the left figure are groups of activated vertices.

We define an abstract pyramid $pyramid(l)$ as an activated diagram without an initial vertex or an initial time. In $pyramid(l)$, we always consider that the vertices at the bottom are primary ones. Then, $pyramid(l)$ turns into $pyramid(l, i, t)$ if placed in a base diagram (i.e., if we choose an initial vertex $i$ and a time $t$).

Consider a chinampa $C$. Let **P** be the set of $pyramid(l)$, with one for each stacking of $pyramid(l, i, t)$ in a chinampa $C$. We recover the chinampa by stacking abstract pyramids from the set **P**. We associate the corresponding pyramidion with any $P \in$ **P**. If the pyramidion of $P$ belongs to the pyramid $Q$, then we call $Q$ the *parent* of the pyramid $P$. Note that after stacking, some primary vertices of the parent become secondary vertices.

We describe an algorithm to create the list **P**. We can use the breadth-first search (BFS) or depth-first search (DFS) algorithm [17]. Algorithm 1 is the pseudocode of an implementation of BFS. In this algorithm, the pyramidion and the parent of $P$ are attached to $P$ as $P.pyramidion$ and $P.parent$, respectively.

---

**Algorithm 1:** Factorization via BFS

**Data:** A chinampa
**Result:** A Unique factorization
$Pyramid_0$ = pyramid whose pyramidion is the spike.;
Cache = $[Pyramid_0]$;
factorization = [ ];
**while** *Cache != [ ]* **do**
    init = pop(Cache);
    `/* compare the vertices of init with pyramidion        */`
    **for** *v in V(init)* **do**
        **if** *v==init.pyramidion* **then**
            continue;
        **else if** *there is P $\in$ **P** with P.parent == init and P.pyramidion == v* **then**
            Add $P$ to Cache
    **end**
    factorization.append(init)
**end**
**return** *factorization*

---

**Remark 5.** *The vertices in a pyramid are ordered. For example, the dictionary order starts at the top and proceeds from left to right. Provided a fixed order for abstract pyramids, as described above, our algorithm is deterministic. Thus, we consider the returned list as a factorization of the chinampa in terms of pyramids.*

## 5. Cascades and Cellular Atomata

We adopted the convention from [18], stating that "the network, node, link combination often refers to real systems ... In contrast, we use the terms graph, vertex, edge when we discuss the mathematical representation of these networks".

### 5.1. Base Diagrams as a Model of a Network

Our networks are such that there exist nodes $p$ firing due to external stimuli and nodes $s$ firing when the sum of the input signals from other nodes exceed their thresholds. Our goal is the study of cascades, which are networks whose number of nodes $s$ is greater or equal to the number of nodes $p$. We translated the networks to a base diagram $(B, P : B \mapsto A)$, where a node $p$ firing due to external stimuli at time $t$ in the network corresponded to a primary vertex $(p, t)$ of $V(B)$, and a node $s$ firing as a consequence of the signals of other nodes corresponded to a secondary vertex in $V(B)$. In this way, a cascade in a network corresponded to an activation diagram. The activation diagram in Figure 3 is associated to a cascade.

Table 1 resumes the relation between the definitions of networks and the corresponding definitions of the base diagrams. The terms on the left relate to network theory, and the terms on the right are the equivalent concepts in the base diagrams.

**Table 1.** Equivalent definitions between networks and base diagrams.

| Network Theory | Base Diagram |
|---|---|
| External stimulated node | Primary vertex |
| Internal stimulated node | Secondary vertex |
| (Network, external stimuli) | Activation diagram (AD) |
| Cascade | AD with equal or more secondary vertices than primary ones |

### 5.2. Cellular Automata

As we will see in this subsection, a cascade can be interpreted as a cellular automaton. We considered cellular automata [19] in which three consecutive colored (black or white) cells determined the color of the middle cell in the next iteration. In particular, we were interested in rule 192. This rule dictates that cell $C$ will be black if $C$ and the one to the left were black in the previous iteration (see Figure 12).

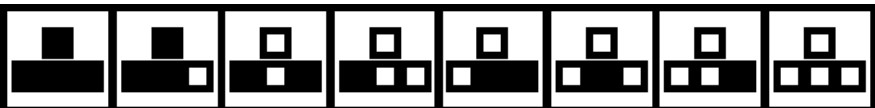

**Figure 12.** Rule 192. Only the first and second stages lead to a black cell, similar to our hypothesis on the signal flow graphs.

In Figure 13, the vertical axis is the time, and we thought of the yellow blocks as the stimuli to keep alive the cells of the game of life [20]. The lowest row represents a cell which survived for three units of time under rule 192. The next row shows 3 automata which survived 4 units of time, 6 which survived 5 units of time, and 12 which survived 6 units of time.

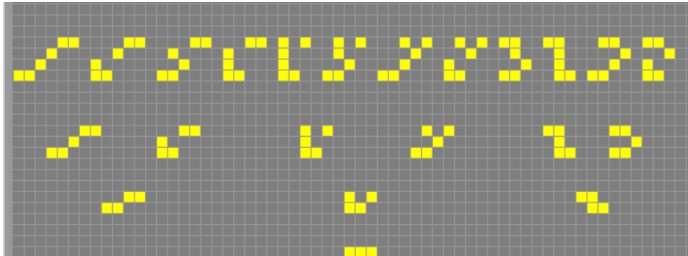

**Figure 13.** Cellular automata with less than six external stimuli.

Thus, a cascade on networks can be seen as the evolution of cellular automata, where we fixed the external stimuli and applied rule 192 to evolve the cells. More specifically, we employed rule 192 [19] but allowed the initial conditions to occur at different times. Accordingly, we could use the language of graph theory to describe the behavior of the cellular automata (cascade).

## 6. Combinatorial Description of Chinampas

Since a base diagram is a set of chinampas, we studied the properties of chinampas. As a part of our contributions, we present some combinatorial properties of a chinampa. For example, we will give a formula for the number of chinampas inside a pyramid. This formula is given in terms of a generating function. Another result is the prediction of when a vertex appears as a secondary vertex in a chinampa. This can be accomplished by using algorithms to find the pyramids in a chinampa.

### 6.1. Profit Properties

In the current context, since there is no confusion between a pyramid and an abstract pyramid, we sometimes refer to both as pyramids. From now on, we will construct chinampas via the stacking process. Remember that we do not allow the stacking of two pyramids when one ends inside the other, neither one is adjacent to the other, or some primary vertices in the abstract pyramid turn into secondary vertices once we stack a pyramid.

Let $P_1$ and $P_2$ be two abstract pyramids, and let $C$ be the chinampa obtained by stacking $P_1$ into $P_2$. We define the *intersection $P_1 \cap P_2$* of two abstract pyramids $P_1$ and $P_2$ as the abstract pyramid with activated vertices at the intersection of the activated vertices of $P_1$ and $P_2$. Given the definition of an abstract pyramid, if a vertex is primary in one vertex and secondary in the other, then the vertex is primary in $P_1 \cap P_2$.

Remember that we defined the profit as the difference between the secondary vertices and the number of primary vertices. If the intersection of two pyramids is only one point, hen we assume a profit of $-1$. We have the following result for vertical stacking:

**Lemma 1.** *Let $C$ be a chinampa obtained by vertical stacking of two abstract pyramids $P_1$ and $P_2$. The profit function satisfies*

$$profit(C) = profit(P_1) + profit(P_2) - profit(P_1 \cap P_2).$$

**Proof.** Suppose pyramid $P_1$ is above $P_2$. Then, the abstract pyramid $P_3 = P_1 \cap P_2$ is once again a pyramid. Let $n_1$, $n_2$, and $n_3$ be the number of primary vertices of $P_1$, $P_2$, and $P_3$, respectively. Then, the $n_3$ primary vertices of $P_1$ turn into secondary vertices in $C$, and therefore the number of primary vertices of $C$ is given by

$$(n_1 - n_3) + n_2,$$

What is more, let $m_1$, $m_2$, and $m_3$ be the number of secondary vertices of $P_1$, $P_2$, and $P_3$, respectively. Then, the number of secondary vertices of $C$ is

$$(m_1 + n_3) + m_2 - (m_3 + n_3).$$

Therefore, the results follow. □

Note that the arguments in the proof of Lemma 1 are standard and can be used to prove the following proposition:

**Proposition 3.** *Let* $C$ *be a chinampa with a set of abstract pyramids* **P***. The profit function satisfies the inclusion-exclusion principle*

$$profit(C) = \sum_{P \in \mathbf{P}} profit(P) - \sum_{P_1, P_2 \in \mathbf{P}} profit(P_1 \cap P_2) + \sum_{P_1, P_2, P_3 \in \mathbf{P}} profit(P_1 \cap P_2 \cap P_3) \pm \cdots . \quad (1)$$

There are two approaches to stacking one $pyramid(2)$ into another $pyramid(2)$. According to Equation (1), stacking $pyramid(2)$ into a chinampa does not change the profit. We describe how the action of stacking affects the profit:

**Lemma 2.** *Let* $C$ *be a chinampa with profit n. Stacking a* $pyramid(l, i, t)$ *with* $l \geqslant 3$ *into* $C$ *creates a chinampa* $C'$ *with profit* $m > n$.

**Proof.** Assume that we stack $pyramid(l, i, t)$ into $C$. Locally, $pyramid(l, i, t)$ is stacked in a pyramid $P$ of $C$ so that the primary vertices of $P$ with time $t + s$ where $0 < s < l$ are replaced by secondary vertices (see Figure 14). The number of primary vertices of $C$ increases by $l - (l - s) = s$, although the number of secondary vertices increases by $(l - 1 + l - 2 + ... + l - s)$ because stacking only affects the vertices of $P$. The result follows since $l \geqslant 3$ implies $(l - 1 + l - 2 + ... + l - s) > s$. □

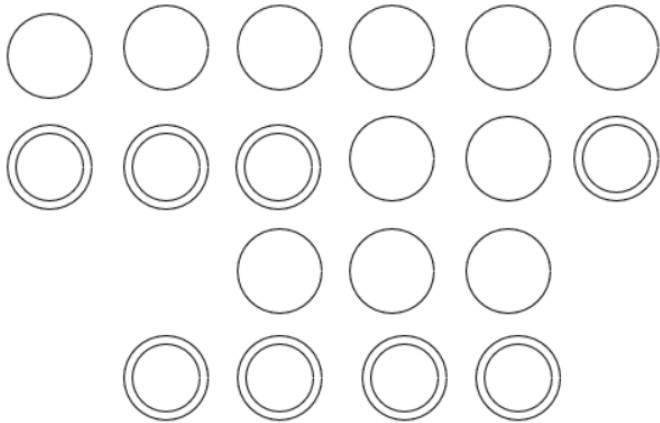

**Figure 14.** Here, $pyramid(l, i, t)$ with $l = 4$ is stacked into a chinampa $C$. In this example, the two primary vertices of $C$ at $t + 2$ become secondary.

**Remark 6.** *Stacking one pyramid with a length of three and several pyramids with a length of two returns a chinampa with a profit of zero. Conversely, under Lemma 2, if we have a chinampa with a profit of zero, then it must be the result of stacking one pyramid with a length of three and some pyramids with a length of two. Similarly, a chinampa with a profit of one has two copies of pyramids with a length of three and several copies of pyramids with a length of two. For a profit of two, we can have either one* $pyramid(4)$ *and several* $pyramid(2)$ *instances stacked on or below it or three* $pyramid(3)$ *instances and several* $pyramid(2)$ *instances.*

In general, a chinampa with a profit $k$ is made by stacking several $\{pyramid(l_i)\}$ so that they satisfy Equation (1). We know that $profit(pyramid(l)) = \frac{l(l-3)}{2}$, which is less than or equal to $k$ under Lemma 2. This gives a bound on the largest pyramid contained in a chinampa in terms of the profit of the chinampa

$$l_i \leqslant \frac{3 + \sqrt{9 + 8k}}{2}, \tag{2}$$

for all $l_i$.

### 6.2. Combinatorial Description of Chinampas with Profits of Zero and One

We aim to find the number of chinampas inside $pyramid(n)$. A general chinampa can always be considered as part of $pyramid(n)$ (see Remark 4). Therefore, consider $P = pyramid(n)$ in $tellis(n)$. We define

$$ch[n; (2, a_2), (3, a_3), \ldots, (k, a_k)]$$

to be the number of all chinampas contained in $P$ and obtained by stacking $a_i$ copies of $pyramid(i)$, where $2 \leqslant i \leqslant k$.

**Example 2.** *It is clear that* $ch[n; (n, 1)] = 1$, $ch[n + k; (n, k + 1)] = 2^k$ *for* $k \geqslant 0$ *and*

$$ch[n; (2, n - 1)] = 2^{n-2} \tag{3}$$

To simplify the notation, from now on, we will omit the terms corresponding to $pyramid(2)$, although they remain a part of the calculations. Thus, $ch[n; (2,1), (n-1,1)]$ becomes $ch[n; (n-1,1)]$.

In Figure 15, we show three of the elements identified by $ch[4; (3,1)]$. The three chinampas with a profit of zero have $pyramid(3)$ at the top.

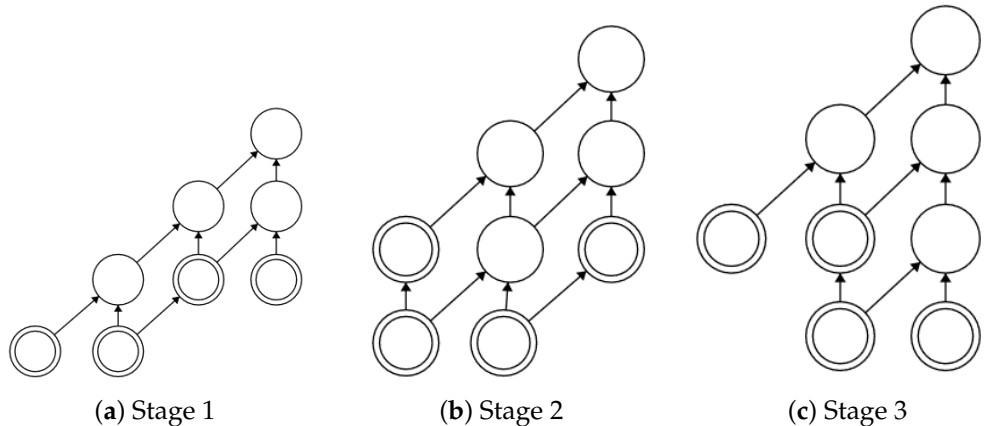

(**a**) Stage 1         (**b**) Stage 2         (**c**) Stage 3

**Figure 15.** Three chinampas with a profit of zero having $pyramid(3)$ at the top. Note that the stages were obtained from $pyramid(3)$, replacing one of the three primary vertices $v$ at a time of zero by two primary vertices at the time $-1$ so that $v$ becomes a secondary vertex.

To count the number of zero-profit chinampas within an instance of $pyramid(n)$, we consider the formal series

$$\sum_n ch[n + 3; (3,1)] \frac{x^n}{n!}.$$

Following [21], we use the calculus of formal exponential generating functions to determine all coefficients. Note that $ch[n + 3; (3,1)] = 0$ for $n < 0$.

We will use the fact that

$$p(x) = 2 \int p(x) dx + f(x) e^{2x} + h(x)$$

has the solution

$$p(x) = \frac{d}{dx}\left(e^{2x}\int f(x) + h(x)e^{-2x}dx\right).\tag{4}$$

**Proposition 4.** *In pyramid$(n+3)$, for the nonnegative integer $n$, there are $(2+3n)2^{n-1}$ possible zero-profit chinampas. Furthermore, these numbers are given by the coefficients of the generating function*

$$\sum_{n=0}^{\infty} ch[n+3;(3,1)]\frac{x^n}{n!} = (3x+1)e^{2x}.$$

**Proof.** Recall that according to Remark 6, a zero-profit chinampa has only one stacked *pyramid*(3). First, we can explicitly count the number of chinampas in *pyramid*$(n+3)$ when *pyramid*(3) is at the top of *pyramid*$(n+3)$. For each integer $n \geqslant 0$, there are $3(2^{n-1})$ such possible chinampas. This is because for each element in Figure 15, we create new elements by stacking a sequence of *pyramid*(2) below the element. Therefore, the result follows from Equation (3). The remaining chinampas in *pyramid*$(n+3)$ are those for which *pyramid*(3) is not at the top, namely $2ch[n-1+3;(3,1)]$. This follows because *pyramid*(3) is within one of the two subpyramids *pyramid*$(n-1+3)$: one given by ignoring the main diagonal of *pyramid*$(n+3)$ or the other by ignoring the right vertical column of *pyramid*$(n+3)$.

Therefore, for $n > 0$, we have

$$ch[n+3;(3,1)] = 2ch[n-1+3;(3,1)] + 3(2^{n-1}).$$

Now, we define

$$p(x) = \sum_{n=0}^{\infty} ch[n+3;(3,1)]\frac{x^n}{n!}.$$

so

$$p(x) = \sum_{n=0}^{\infty} ch(n+3,(3,1))\frac{x^n}{n!}$$

$$= 1 + 2\sum_{n=1}^{\infty} ch(n-1+3,(3,1))\frac{x^n}{n!} + \sum_{n=0}^{\infty} 3(2^{n-1})\frac{x^n}{n!}$$

$$= 2\int p(x)dx + \frac{3e^{2x}-1}{2}$$

Using Equation (4) and the condition $ch[3;(3,1)] = 1$, we obtain

$$\sum_{n=0}^{\infty} ch[n+3;(3,1)]\frac{x^n}{n!} = (3x+1)e^{2x}$$

$$= \sum_{n=0}^{\infty} (2+3n)2^{n-1}\frac{x^n}{n!}$$

$\square$

**Proposition 5.** *Chinampas of a certain unit of profit have the generating function*

$$\sum_{n=0}^{\infty} ch[n+4;(3,2)]\frac{x^n}{n!} = (9x^2 + 18x + 4)\frac{e^{2x}}{2}.$$

**Proof.** A chinampa unit of profit can only be formed by two copies of *pyramid*(3) and chains of *pyramid*(2), as shown in Remark 6. We define the formal series

$$q(x) = \sum_{n=0}^{\infty} ch[n+4; (3,2)] \frac{x^n}{n!},$$

and consider the following cases:

- None of the instances of *pyramid*(3) are at the top. We then have subpyramids as in the previous proposition, so we count $2ch[n+3; (3,2)]$.
- One *pyramid*(3) instance is at the top. Then, the remaining *pyramid*(3) instances can be placed in $2ch[n+3; (3,1)]$ ways on the two subpyramids. However, the two subcases have $ch[n+2; (3,1)]$ terms in common, as shown in Figure 16. Thus, the correct number of combinations is $2ch[n+3; (3,1)] - ch[n+2; (3,1)]$.

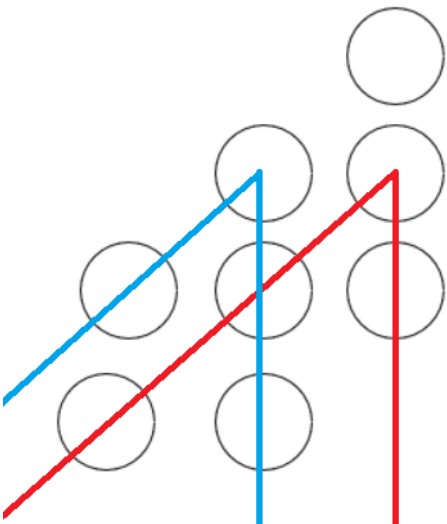

**Figure 16.** Stacking any pyramid onto pyramid(3) in the vertex, where the red and blue lines collide and continue the process of stacking pyramids iteratively. This process describes a family of pyramids counted twice: once under the blue region and once under the red region.

We conclude that for each nonnegative integer $n$, we have

$$ch[n+4; (3,2)] = 2ch[n+3; (3,2)] + 2ch[n+3; (3,1)] - ch[n+2; (3,1)].$$

Then, using the generating series $p(x)$ found in Lemma 4, we compute

$$
\begin{aligned}
q(x) &= \sum_{n=0}^{\infty} ch[n+4; (3,2)] \frac{x^n}{n!} \\
&= 2\sum_{n=1}^{\infty} ch[n+3; (3,2)] \frac{x^n}{n!} + 2\sum_{n=0}^{\infty} ch[n+3; (3,1)] \frac{x^n}{n!} - \sum_{n=1}^{\infty} ch[n+2; (3,1)] \frac{x^n}{n!} \\
&= 2\int q(x)dx + 2(3x+1)e^{2x} - \int (3x+1)e^{2x} dx \\
&= 2\int q(x)dx + \frac{18x+9}{4} e^{2x}
\end{aligned}
$$

By solving Equation (4), we obtain

$$\sum_{n=0}^{\infty} ch[n+4; (3,2)] \frac{x^n}{n!} = \frac{9 + 2c + 36x + 18x^2}{4} e^{2x}$$

We compute $ch[4; (3,2)] = 2$ to conclude

$$\sum_{n=0}^{\infty} ch[n+4;(3,2)]\frac{x^n}{n!} = (4+18x+9x^2)\frac{e^{2x}}{2}.$$

$\square$

### 6.3. Algorithms for Activated Vertices

We will describe an algorithm to construct stacked pyramids of chinampas and to predict whether a vertex is activated at a given time within a chinampa.

Building Chinampas

Following ideas from dynamical programming, we compute the pyramids that make up a chinampa. As a reminder, we assume that each $pyramid(l)$ comes with a natural time ordering on its vertices.

We first construct a chinampa with a known set of primary vertices sorted by their time of occurrence, followed by a secondary sorting of the vertex labels. For every sequence of $n$ consecutive primary vertices with the same time $t$, we assign a new structure named an *interval*. An interval only remembers the coordinates $(n_0, t), (n_1, t)$ of the first and last primary vertices, while $t$ is the time of the interval's occurrence. We assign the order inherited from the set of primary vertices to the set of intervals. The left vertex of the interval establishes the order.

Next, we associate $pyramid(l)$ with the interval with the lowest parameter $t$ that has $l$ consecutive vertices. An iterative process to build the chinampa is as follows. Given the next interval, defined by consecutive vertices $l_k$ with $t' \geqslant t$, we check whether the first or last term is next to a pyramid built previously. If so, then we extend the interval to include the secondary vertices of the pyramid, whose time equals $t'$. Once we grow the interval from $l_k$ to $l_k + d_k$, we assign to the interval the $pyramid(l_k + d_k)$ (see Algorithm 2). The auxiliary Algorithm 3 removes duplicates.

Let $n$ be the number of primary vertices, and let $n_0 = int(n/2)$. To compute the time complexity of Algorithm 2, we analyzed the best-case and worst-case scenarios. The best-case scenario is where all vertices are part of the base of a pyramid. In the best-case scenario, the computational complexity is $O(n)$. The worst-case scenario is where we have $n_0$ copies of $pyramid(n_0)$ concatenated such that two consecutive pyramids with different times share the maximum area possible. In the worst-case scenario, the algorithm complexity is $O(n^2)$ due to the double loop.

To determine whether a vertex is activated, we must determine whether it is contained within a pyramid. Therefore, for each pyramid, one must verify whether the vertex satisfies the constraints necessary to keep them within the region defined by the corners of the pyramid. See Algorithm 4 for the time complexity $O(n)$ in the best case and $O(n^2)$ in the worst case (because it calls back to Algorithm 2). The source code of this algorithm can be accessed at https://github.com/mendozacortesgroup/chinampas/ (accessed on 3 March 2023).

---

**Algorithm 2:** Build chinampas

---

**Input:** An ordered list of primary vertices *PV*.
**Result:** A list of pyramids;
listOfPyramids = [ ];
interval.t= PV[0].time, interval.lP=PV[0].position,
    interval.rP=PV[0].position;// time, left pos,right pos
**for** *vertex in PV*[1 :] **do**
   **if** *vertex.time == interval.t and vertex.position = interval.rP+1* **then**
     | interval.rP = vertex.position
   **else**
     | listOfPyramids.append(interval);
     | interval.t=vertex.time, interval.lP=vertex.position,
     |  interval.rP=vertex.position;
   **end**
**end**
listOfPyramids.append(interval);
pastPyramids = copy(listOfPyramids);
**for** *interval in ListOfPyramids* **do**
   **for** *lowerPyramid in pastPyramids* **do**
     | deltaT=(interval.t-lowerPyramid.t);
     | **if** *lowerPyramid.t+(lowerPyramid.rP-lowerPyramid.lP) <interval.t* **then**
     |  | pastPyramids.remove(lowerPyramid)
     | **else if** *lowerPyramid.t>interval.t +interval.rP-interval.lP* **then**
     |  | break
     | **else if** *interval.lP==lowerPyramid.rP+1* **then**
     |  | interval.lP=lowerPyramid.lP+deltaT
     | **else if** *interval.rP==lowerPyramid.lP+deltaT-1* **then**
     |  | interval.rP=lowerPyramid.rP
     | **end**
   **end**
**end**
listOfPyramids = removeDuplicates(listOfPyramids); // see Algorithm 3
**return** listOfPyramids

---

**Algorithm 3:** Remove duplicates

---

**Input:** An ordered list of intervals *listOf Pyramids*;
**Result:** An ordered list of intervals without intersection;
dynamicCopy=copy(listOfPyramids);
previous = dynamicCopy[0];
index = 1;
**for** *walker in listOf Pyramids*[1 :] **do**
   **if** *previous.rP==walker.lP-1 and previous.t==walker.t* **then**
     | previous.rP=walker.rP
     |  dynamicCopy=dynamicCopy[:index]+dynamicCopy[index+1:];
   **else**
     | previous =dynamicCopy[index];
     | index = index+1;
   **end**
**end**
**return** *dynamicCopy*

---

**Algorithm 4:** Will_vertex_be_activated

---

**Input:** A vertex $(n, t_0)$, a list of primary vertices $PV$.
**Result:** Boolean explaining if the vertex $n$ will be activated at time $t_0$;
**if** $(n, t_0)$ *in PV* **then**
  **return** True ;
**end**
orderedListOfPyramids = $buildPyramids(PV)$;
**for** *pyramid in orderedListOfPyramids* **do**
  deltaT=$(t_0 - pyramid.t)$;
  **if** *pyramid.lP + deltaT $\leqslant$ n $\leqslant$ pyramid.rP and*
  *pyramid.t $\leqslant$ $t_0$ $\leqslant$ n $-$ pyramid.lP + pyramid.t* **then**
    /* $(n, t_0)$ in a pyramid                    */
    **return** True;
  **else if** *pyramid.t $>$ $t_0$* **then**
    break;
  **end**
**end**
**return** False;

---

## 7. Triangular Sequences

We study the chinampas obtained by stacking several *pyramid*(2) instances below *pyramid*(n) for $n \geqslant 4$. For ease of exposition, we define *roots* as the sequences of *pyramid*(2) stacked on top of each other. Note that *pyramid*(n) with $n \geqslant 4$ can have multiple roots. For simplicity, we let *pyramid*(n) have $n = 3K$ and $K \in \mathbb{N}$, and all roots had the same number $R$ of *pyramid*(2).

**Example 3.** *Consider the two extreme cases for pyramid(6), whose roots are depicted in Figure 17. In Figure 17a, the roots are formed by stacking pyramid(2) vertically, while in Figure 17b, the pyramid(2) instances are stacked along diagonals.*

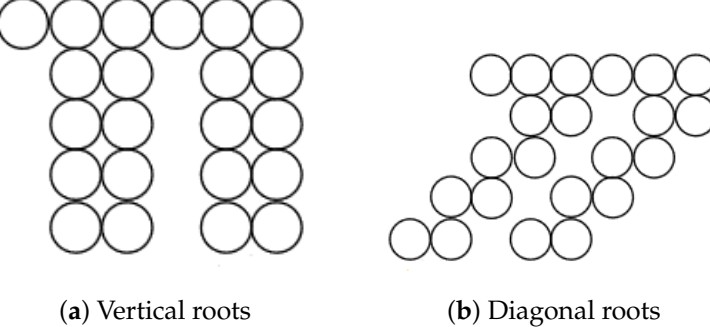

(**a**) Vertical roots                    (**b**) Diagonal roots

**Figure 17.** Roots in *pyramid*(6). We show all activated vertices except those above the primary vertices of *pyramid*(6).

Given $K$ and $R$, we define a *KR-triangular sequence* as a sequence with two indexes $\{s_i^j\}$, where the coefficients are integers and they satisfy the constraints

$$
\begin{array}{ccccccccccc}
K+R & & K+R-1 & & & & K+i & & & K+2 & K+1 \\
\vee\!\!/ & & \vee\!\!/ & & & & \vee\!\!/ & & & \vee\!\!/ & \vee\!\!/ \\
s_R^R & > & s_{R-1}^{R-1} & > & \cdots & > & s_i^i & > & \cdots & > & s_2^2 & > & s_1^1 \\
\vee\!\!/ & & \vee\!\!/ & & & & \vee\!\!/ & & & \vee\!\!/ & \vee\!\!/ \\
R & & s_R^{R-1} & > & \cdots & > & s_{i+1}^i & > & \cdots & > & s_3^2 & > & s_2^1 \\
& & \vee\!\!/ & & & & \vee\!\!/ & & & \vee\!\!/ & \vee\!\!/ \\
& & R-1 & & & & \vdots & & & \vdots & \vdots \\
& & & & & & \vee\!\!/ & & & \vee\!\!/ & \vee\!\!/ \\
& & & \cdots & > & & s_R^i & & \cdots & > & s_{j+1}^2 & > & s_j^1 \\
& & & & & & \vee\!\!/ & & & \vee\!\!/ & \vee\!\!/ \\
& & & & & & i & & & \vdots & \vdots \\
& & & & & & & & & \vee\!\!/ & \vee\!\!/ \\
& & & & & & & & & s_R^2 & > & s_{R-1}^1 \\
& & & & & & & & & \vee\!\!/ & \vee\!\!/ \\
& & & & & & & & & 2 & & s_R^1 \\
& & & & & & & & & & & \vee\!\!/ \\
& & & & & & & & & & & 1
\end{array}
\tag{5}
$$

**Remark 7.** *The particular relation $s_j^i > s_{j-1}^{i-1}$ prevents redundancy of the roots.*

**Proposition 6.** *Consider a chinampa with multiplicity roots $n = 3K$, where each root with $R$ copies pyramid$(2)$. The number of KR-triangular sequences $\{s_j^i\}$ counts the number of possible roots on pyramid$(n)$ with the previous conditions.*

**Proof.** Given the *KR*-triangular sequence $\{s_j^i\}$, consider a rectangular board $B$ of $(K+R)$ columns and $R$ rows.

Step 1: We place a white mark at the cell of $B$, given by the intersection of row 1 and column $s_1^1$. Step 2: We place a white mark at the cell intersection of row 2 and column $s_2^1$ and another white mark at the cell intersection of row 2 and column $s_2^2$. Step $i$ requires us to place marks at cells $(i, s_j^i)$ with $j \leqslant i$. Now, for each $i$, we take the $i^{th}$ row and color all non-white cells in black from columns 1 to $n + i$.

To recover the roots of *pyramid*$(n)$, we substitute each black cell with three consecutive cells: one white and two black. The black cells are the activated vertices of the roots of *pyramid*$(n)$. The fact that each of the sequences $\{s_j^i\}_j$ is decreasing translates into a movement of the roots to the left. The condition $s_j^i > s_{j-1}^{i-1}$ appears because the roots can move only one unit to the left. The map from one black block to a white block with two black blocks prevents redundancy. This assignment can be verified to be an isomorphism. $\square$

**Example 4.** *We examine the white spaces as shown in Figure 18. They correspond to the roots of Figure 17. According to Proposition 6, the sequences corresponding to Figure 18a are*

$$
\begin{array}{ccccccc}
& & & s_1^1 & & & 3 \\
& & s_2^2 & s_2^1 & & 4 & 3 \\
& s_3^3 & s_3^2 & s_3^1 & = & 5 & 4 & 3 \\
s_4^4 & s_4^3 & s_4^2 & s_4^1 & & 6 & 5 & 4 & 3
\end{array}
$$

*For Figure 18b, they are*

$$
\begin{array}{ccccccc}
& & & s_1^1 & & & 1 \\
& & s_2^2 & s_2^1 & & 2 & 1 \\
& s_3^3 & s_3^2 & s_3^1 & = & 3 & 2 & 1 \\
s_4^4 & s_4^3 & s_4^2 & s_4^1 & & 4 & 3 & 2 & 1.
\end{array}
$$

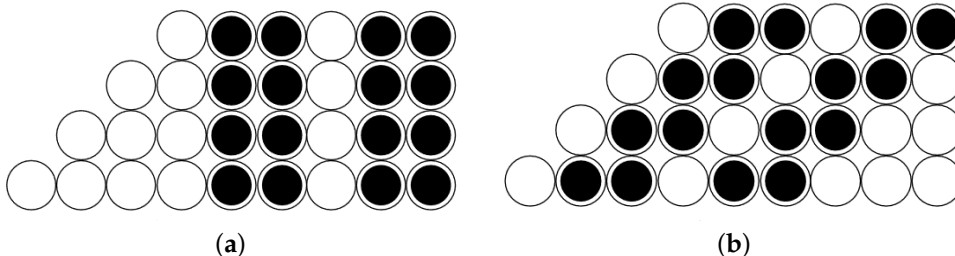

**Figure 18.** The corresponding triangular sequences come from counting blocks. (**a**) Vertical roots. (**b**) Diagonal roots.

*Ehrhart Series and Order Polynomials*

The order polynomial $\Omega(P, x)$ of a partially ordered set (poset) $P$ was introduced by Stanley [22]. The polynomial evaluated on $n$ returns the number of labels $\Omega(P, n)$ on the poset $P$, which preserves the order.

Similar to the construction of $T_{R,n}$, we can construct a poset $P_R$ containing only the symbol $\geqslant$ by subtracting $i - 1$ units from the column $i$ from right to left, where $i \in [1, \cdots, R]$. The poset only depends on the variable $R$ and not on the variables $k$ and $n$:

**Lemma 3.** *Consider chinampas with roots of multiplicity $n = 3K$, where each root with R is a copy of pyramid(2). Then, the number of such chinampas is $\Omega(P_R, k + 1)$.*

**Proof.** The number of triangular sequences $T_{R,n}$ is $\Omega(P_R, k + 1)$. $\square$

This connection with combinatorics allowed us to determine the properties of the generating functions. For a poset $P$, one can associate the order polytope [23,24] $Poly(P)$. Then, the generating function of the order polynomial is the variable $x$ times the Ehrhart series. For example, when $R = 3$, the triangular sequence corresponds to the poset $\{a < b < c < d < e, b < f < d\}$, and we obtain the generating function $\frac{-x}{(1-x)^6} + 2\frac{x}{(1-x)^7}$.

Ehrhart series of order polytopes are known to be of the form $\frac{h^*(x)}{(1-x)^{d+1}}$, or in our case $d = \frac{R(R+1)}{2}$. The term $h^*(x)$ is a polynomial of a degree of at most $d$, where its coefficients satisfy the Dehn–Sommerville equations and are unimodal.

The previous result relates the enumeration of chinampas with polytopes of the form $Poly(P_R)$.

We counted chinampas with the help of Mathematica [25] and a topological version of the calculus of species [26]. These calculations gave us evidence of the validity of Lemma 3 and led to the concept of triangular sequences. For details on the use of Mathematica for counting order polytopes, see [27].

## 8. Conclusions

In this paper, we introduced activation diagrams, chinampas, and pyramids to study the effect of signals on the vertices of a nonlinear signal flow path. Furthermore, we demonstrated that pyramids are the simplest possible activation diagrams. Finally, we presented a deterministic algorithm to construct chinampas out of pyramids (see Remark 5). Chinampas were conceived to serve as an idealized model for cascades, with sequences of neural spikes forming a polychrony group.

To support our conclusions, we developed an optimal code to answer the following introductory questions: "Will a fixed vertex be activated at a particular time? Can we reconstruct all the vertices that will be activated?" We also achieved the enumeration of chinampas of profits of zero and one. The problem of finding all chinampas of profits bigger than two remains open. Our techniques for counting the chinampas of profits of zero and one cannot be adapted to this case, as the techniques miss a large family of elements (see Remark 6). We established that for a family of chinampas represented by triangular

sequences, their enumeration problem is equivalent to computing the Stanley-order polynomial of a family of posets. To the best of our knowledge, finding the order polytope or Erhart series of these posets remains an open problem in enumerative combinatorics.

Modern algorithms aim to emulate the behavior of brain regions by simulating polychrony groups across multiple neurons [28,29]. Our approach differs in that we focus on a particular path of neurons and study the possible polychrony groups on that network.

Our contribution to computational neuroscience is not only theoretical. For example, the algorithm included in [1] and the software of [16,30–33] each emulate multiple cascades in parallel. When studying individual cascades along a line or in a tree, the software evaluates each cascade with a computational complexity of $O(n^2)$, where $n$ is the number of neurons. In comparison, our algorithms scale as $O(n)$ and $O(n^2)$ in the best and worst cases, respectively. Optimization is important, since brain-like hardware is known to perform poorly [10,34]. We believe that our code can help to better understand the patterns of large polychrony groups efficiently.

Our work is limited to the study of signal flow paths. Possible continuations of this work include modeling triple-spike timing-dependent plasticity [35–38], for example in this paper we used self edges to model long-term potentiation. Inspired by [39], a machine learning algorithm such as genetic algorithms, combined with our software, should produce an algorithm with input of experimental measurements of spikes and output of the most likely topology of the network. Another possibility is to study redundant polychrony groups, where redundancy is applied to assure that software will work even if some components are damaged. Following [40], we would like to introduce noise in the theory of chinampas. Perhaps polycrhony groups can be used to study sparse neural networks [41–45] when the neuronal network uses a sigmoid activation function which is equivalent to our nonlinearity condition for the signal flow graphs. In relation to the theory of species [21,46], the first and second author are currently developing a topological version of species [26]. Topology is needed because our generating functions are parameterized by posets, as in Lemma 3. Finally, we believe it may be of interest to study signal flow graphs that admit cycles according to Figure 2. The feedback enables the existence of perpetual chinampas, the feedback in [14] is used to encode messages.

**Author Contributions:** Conceptualization, E.D.-C., J.A.A.-N., J.L.M.-C. and N.C.; methodology, E.D.-C. and J.A.A.-N.; software E.D.-C. and L.V.P.; validation, E.D.-C., J.A.A.-N., A.N. and A.Y.Z.; formal analysis, E.D.-C., J.A.A.-N. and G.E.; investigation, E.D.-C. and J.A.A.-N.; resources, J.L.M.-C.; writing—original draft preparation, E.D.-C., J.A.A.-N and J.L.M.-C.; writing—review and editing, E.D.-C., J.A.A.-N., N.C., G.E. and J.L.M.-C.; visualization, E.D.-C. and J.A.A.-N.; supervision, J.L.M.-C.; project administration, E.D.-C. and J.L.M.-C.; funding acquisition, G.E. and J.L.M.-C. All authors have read and agreed to the published version of the manuscript.

**Funding:** The first author received funding from a National Research Foundation of Korea (NRF) grant funded by the Korean government (MSIT) (No. 2020R1C1C1A01008261). We thank FSU open access fund for covering publication costs.

**Acknowledgments:** We thank Jade Master for clarification of the relationship between Petri nets and chinampas. We thank the reviewers for their comments and suggestions that helped to improve this work. The cellular automata drawings were made using http://madebyevan.com/fsm/ (accessed on 3 March 2023). Figure 13 was made with https://playgameoflife.com/ (accessed on 3 March 2023).

**Conflicts of Interest:** The authors declare no conflict of interest.

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
