# Peer review of "Polychrony as Chinampas"

_algorithms, doi:10.3390/a16040193_

Round 1

Reviewer 1 Report

This paper studies the signal-flow graphs. The paper must be thoroughly improved. Specifically, the following concerns need to be addressed.

1.      The abstract is too brief, and fails to summarize the paper properly.

2.      What are the limitations of the previous works? The motivation of the paper must be further clarified.

3.      The subfigures need to be numbered properly (in accordance with the corresponding template). In addition, each subfigure must be accompanied with its own caption.

4.      Many figures must be explained in more details.

5.      It would be better if the concluding remark can be made at the end of the paper.

6.      Most of the references are old ones. The timeliness of the paper is in doubt. The literature review must be improved.

7.      The algorithmic complexity can be evaluated by comparing with the previous works.

Author Response

We want to thank the reviewer for his questions and comments, we have seen a great improvement in the presentation of our work meeting the requirements he has made. We have rewritten the introduction to improve it, included new bibliography both in the introduction and in the conclusions to better support our work, made small improvements in the structure of the article to make it easier to read and better localize the methodology and results, and finally we have included a conclusions section. Here our answers to the particular reviewer's requests:

  1. The abstract is too brief...
    Ans. We have redistributed the article. In particular, we have rewritten the introduction. We believe it is better for the reader. In this introduction we have added the topics covered in each of the sections. We have also added the results we obtained in each section. The new structure of the article also makes it easier to identify the methodology and results.
  2. What are the limitations of the previous works? 
    Ans. In the conclusions section, we have added the difficulties we have encountered in the development of the work, our contributions and additional bibliography to compare our methodology and results with other authors. We also add the limitation of the work and possible continuations.
  3. The subfigures need to be numbered properly...
    Ans. Done.
  4. Many figures must be explained in more details.
    Ans. Done.
  5.  It would be better if the concluding remark can be made at the end of the paper.
    Ans. Done. In the conclusions section, we have taken up a couple of observations (Remarks 5 and 6) that can be considered as conclusions and added more observations about our work. 
  6. Most of the references are old ones...
    Ans. We have added more references in the introduction and emphatically also in the conclusions section. The references are current and closely related to our work.
  7. The algorithmic complexity can be evaluated by comparing with the previous works.
    Ans. Our algorithms contribute to knowledge, as they are an original approach to the problem. In addition, we compare the complexity of our algorithms with that of the existing literature (see references [1, 14, 28 31]). We observed that our algorithms are less complex than those of the existing literature.

The English language was extensively edited.

Reviewer 2 Report

The paper is very well written and concepts are very well illustrated. However, I feel that the paper needs to be improved regarding the following aspcets:

1- The contribution needs further emphasize in the introduction. Currently there is a single sentence.

2- The state-of-the-art is ignored. The paper needs to review existing solutions and algorithms;

3- The paper must provide a comparison between existing solutions and the proposed one;

4- The paper needs to draw conclusions. Currently there is now section dedicated to this aspect. All scientific papers need a conclusion section. Therein, besides conclusions about the proposed algorithms and methodologies, their limitations should be stated. Also, possible future improvements of the presented work should be given.

Author Response

We want to thank the reviewer for his questions and comments, we have seen a great improvement in the presentation of our work meeting the requirements he has made. We have rewritten the introduction to improve it, included new bibliography both in the introduction and in the conclusions to better support our work, made small improvements in the structure of the article to make it easier to read and better localize the methodology and results, and finally we have included a conclusions section. Here our answers to the particular reviewer's requests:

  1. The contribution needs further emphasize in the introduction.
    Ans. We have improved the introduction to include the contributions, in fact, we highlight the achievements in each section. 
  2. The state-of-the-art is ignored. 
    Ans. We have added more and better bibliography in the introduction and, emphatically, also in the conclusions section to contribute to the state of the art.
  3. The paper must provide a comparison between existing solutions and the proposed one.
    Ans. We have now made such a comparison in the fourth paragraph of the conclusions section.

  4. The paper needs to draw conclusions.
    Ans. We have complied with your request adding a conclusions section. Thank you for this, we believe that this point has greatly improved the presentation of our work. We hope that we have complied with each of the points set forth in this request in that section. 

We also improve the English language.

Round 2

Reviewer 1 Report

The following concerns require further improvements.

1.      The abstract is still too brief. Only a single sentence has been added.

2.      The list of references still lacks of the latest works. Please focus on the ones in the last 5 years.

Author Response

Thank you again for your comments. Here our answers:

  1. The abstract is still too brief.
    Ans. We have attended to your request by making a better summarization of the document.
  2. The list of references still lacks of the latest works.
    Ans. We have added some new references in the introduction and in the conclusions section, some of them from the last 5 years: please see [6], [7], [24], [40], [44], [45].
    Most of the articles deal with the linear case. We believe that, for this reason, we have not found enough literature as expected. We can see, for example [7], where the authors treat the nonlinear case by dividing it into two linear ones.

We also reread the article to try to find English mistakes. We hope that this will adequately meet your requirements.

Reviewer 2 Report

The authors have now amended the paper to include answers to the  issues I listed in my first review. I advise on accepting the paper as it is.

Congratulations to the authors on their fine work!

Author Response

Thank you very much

Round 3

Reviewer 1 Report

The concerns have been addressed.